# Point-to-point stabilized optical frequency transfer with active optics

Benjamin P. Dix-Matthews [1,2 ✉], Sascha W. Schediwy[1,2], David R. Gozzard [1,2], Etienne Savalle [3], François-Xavier Esnault[4], Thomas Lévèque[4], Charles Gravestock[1], Darlene D'Mello[1], Skevos Karpathakis [1], Michael Tobar[2] & Peter Wolf[3]

Timescale comparison between optical atomic clocks over ground-to-space and terrestrial free-space laser links will have enormous benefits for fundamental and applied sciences. However, atmospheric turbulence creates phase noise and beam wander that degrade the measurement precision. Here we report on phase-stabilized optical frequency transfer over a 265 m horizontal point-to-point free-space link between optical terminals with active tip-tilt mirrors to suppress beam wander, in a compact, human-portable set-up. A phase-stabilized 715 m underground optical fiber link between the two terminals is used to measure the performance of the free-space link. The active optical terminals enable continuous, cycle-slip free, coherent transmission over periods longer than an hour. In this work, we achieve residual instabilities of $2.7 \times 10^{-6}$ rad$^2$ Hz$^{-1}$ at 1 Hz in phase, and $1.6 \times 10^{-19}$ at 40 s of integration in fractional frequency; this performance surpasses the best optical atomic clocks, ensuring clock-limited frequency comparison over turbulent free-space links.

[1] International Centre for Radio Astronomy Research, The University of Western Australia, Perth, Australia. [2] Australian Research Council Centre of Excellence for Engineered Quantum Systems, The University of Western Australia, Perth, Australia. [3] SYRTE, Observatoire de Paris, Université PSL, CNRS, Sorbonne Université, LNE, Paris, France. [4] Centre National d'Études Spatiales (CNES), Toulouse, France. ✉email: benjamin.dix-matthews@research.uwa.edu.au

Modern optical atomic clocks have the potential to revolutionize high-precision measurements in fundamental and applied sciences[1–7]. The ability to realize remote timescale comparison in situations where fiber links are impractical or impossible, specifically, between ground- and space-based optical atomic clocks[8–22], will enable significant advances in fundamental physics and practical applications including tests of the variability of fundamental constants[23,24], general relativity[25,26], searches for dark matter[27], geodesy[28–34], and global navigation satellite systems[35] among others[36–46]. These efforts build on optical timing links developed for timescale comparison between microwave atomic clocks[47–49], and efforts are underway to develop optical clocks that can be deployed on the International Space Station[50] and on dedicated spacecraft[51].

Similarly, timescale comparisons between mobile terrestrial optical clocks[1,52–55], where one or more mobile clocks are able to be deployed and moved over an area of interest, enable ground tests of general relativity and local geopotential measurements for research in geophysics, environmental monitoring, surveying, and resource exploration.

Comparison of both ground- and space-based clocks, and mobile terrestrial clocks, requires frequency transfer over free-space optical links. Just as with timescale comparison over optical fiber links, free-space frequency transfer should have residual instabilities better than those of the optical clocks. However, atmospheric turbulence induces much greater phase noise than a comparable length of fiber[12,19,56,57]. In addition, free-space links through the turbulent atmosphere must also overcome periodic deep fades of the signal amplitude due to beam wander and scintillation. When the size of the optical beam is smaller than the Fried scale of the atmospheric turbulence, the centroid of the beam can wander off the detector, while in the case where the beam is larger than the Fried scale, destructive interference within the beam (speckle) can result in loss of signal (scintillation) and so loss of timescale synchronization[19,58,59]. These deep fades can occur 10s to 100s of times per second for vertical links between the ground and space, and also on horizontal links on the order of 10 km[12,17].

One method to overcome deep fades of the signal is to transmit a series of optical pulses from an optical frequency comb and compare them with another optical frequency comb at the remote site[21]. While deep fades will result in the loss of some pulses, the time and phase information can be reconstructed from the remaining pulses.

Another method to overcome deep fades is to stabilize the spatial noise caused by atmospheric turbulence by active correction of the emitted and received wave front. In general, tip-tilt correction is sufficient when using apertures that are small compared to the Fried scale as beam wander will dominate the deep fades. For large apertures, the effects of speckle scintillation increase and higher-order corrections using adaptive optics may be necessary. Tip-tilt stabilization of beam wander for comparison of atomic clocks has previously been demonstrated over 12 km with 50 mm scale optics[17] and 18 km with larger 250 mm telescopes[8].

A further practical concern for the deployment of free-space links is the ability of the system to acquire and track a moving object[10,60]. In that case, tip-tilt capability is mandatory, and additionally such a system must be robust while also having as low a size, weight, and power as possible for ease of deployment in spacecraft, airborne relay terminals, or mobile ground segments.

In this work, we describe phase-stabilized optical frequency transfer via a 265 m point-to-point free-space link between two portable optical terminals. Both terminals have 50 mm apertures and utilize tip-tilt active optics to enable link acquisition and continuous atmospheric spatial noise suppression. The terminals are human-portable and ruggedized for daily field deployment to demonstrate the suitability for remote optical timescale comparison. The performance of the phase stabilization system was determined using a separate 715 m, phase-stabilized optical fiber link between the two terminals. The phase-stabilized free-space optical transfer exhibits an 80 dB improvement in phase noise at 1 Hz, down to $2.7 \times 10^{-6}$ rad$^2$ Hz$^{-1}$, compared to the unstabilized optical transmission. The active spatial stabilization used at each terminal is effective at suppressing beam wander caused by the atmospheric turbulence, allowing continuous, cycle-slip and deep-fade free, coherent transmission over periods longer than an hour. The resulting fractional-frequency stability of the phase-stabilized optical transfer reaches $1.6 \times 10^{-19}$ with 40 s of integration. At timescales beyond 100 s, the fractional-frequency stability flattens, which we determine to be caused by unstabilized temperature fluctuations in the uncompensated short fibers in the phase stabilization system.

## Results

**Coherent optical stabilization system**. Figure 1 shows the architecture of the phase stabilization systems, as well as the free-space and fiber links used to compare the phase noise performance.

A 15 dBm optical signal from a 1550 nm NKT Photonics X15 Laser was split and passed into two independent phase stabilization systems, detailed in "Methods". One of these phase-stabilized systems operated over the free-space link, and was used to suppress the phase noise resulting from atmospheric turbulence. The second phase stabilization system operated over an optical fiber that ran underground between the local and remote sites, and was used to measure the performance of the free-space transmission.

Each side of the free-space link also incorporated tip-tilt active optical terminals (detailed in "Methods") that were used to suppress the received optical intensity fluctuations and deep fades caused by beam wander due to atmospheric turbulence. The remote terminal additionally had a bi-directional optical amplifier that amplified the incoming optical signal (typically by ~13 dB) before passing it to the phase stabilization system, and amplified the reflected portion of the signal for transmission back over the link.

The free-space link spanned 265 m between two buildings at the Centre National d'Études Spatiales (CNES) campus in Toulouse, as shown in Fig. 1. The link passed over grass, sparse trees, and roads, and was operated during late winter over the course of 2 weeks. The most favorable conditions were when the sky was overcast and wind speed was low.

**Fully coherent transfer over a true point-to-point link**. Figure 2 shows the measurements for the fiber noise floor (gray) and phase stabilization off (red) cases made with a Microsemi 3120A Phase Noise Test Probe. Phase noise measurements for the phase-stabilized cases with (orange) and without (blue) tip-tilt were obtained using an Ettus X300 Software Defined Radio operating as a continuous IQ demodulator, and are also shown in Fig. 2.

Further discussion of the measurement equipment architecture and choice may be found in the Supplementary Note 1. The phase noise Power Spectral Densities (PSD) found using the Ettus X300 shows good agreement with the Microsemi 3120A within overlapping frequency ranges, as shown in Supplementary Fig. 1.

When the phase stabilization and tip-tilt systems are off, the measured noise is expected to be dominated by atmospheric turbulence. In theory[61], the corresponding PSD is expected to decrease as $f^{-8/3}$ for low frequencies, before dropping sharply as

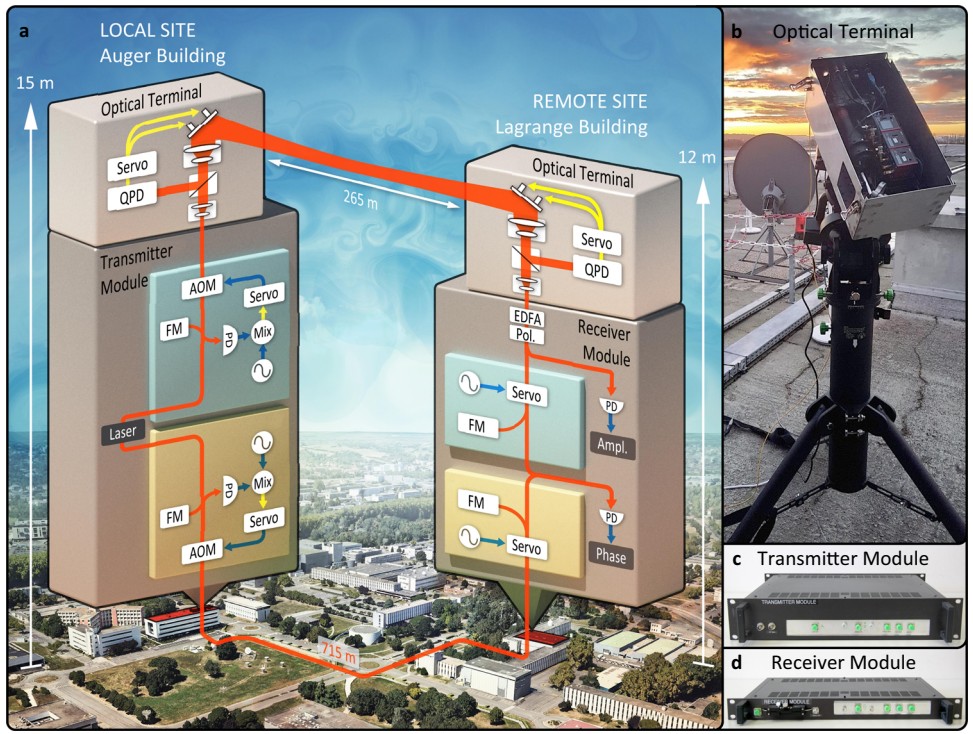

**Fig. 1 Point-to-point phase-stabilized optical frequency transfer between buildings. a** Block diagram of the experimental link. Two identical phase stabilization systems are implemented across the CNES campus. Both systems have their transmitter located in the Auger building (local site), and both receivers are located in the Lagrange building (remote site). One system transmits the optical signal over a 265 m free-space path between the buildings using tip-tilt active optics terminals while the other transmits via 715 m of optical fiber. The relative stability of the two optical signals is then measured at the remote site. QPD, quad-photodetector; Pol, polarization controller; PD, photodetector; PLL, phase-locked loop; AOM, acousto-optic modulator; FM, Faraday mirror; EDFA, erbium-doped fiber amplifier; Mix, radio frequency electronic mixer. Satellite image adapted from Google (Map data: Google, Maxar Technologies). **b** Active optical terminal located at the local site. **c** Transmitter portion of the phase stabilization system located at the local site. **d** Receiver portion of the phase stabilization system located at the remote site.

$f^{-17/3}$ due to the averaging effect of the optical aperture. The slopes of our measured PSD are compatible with that model. The transition frequency between the two regimes is given in ref. [61] by $f_c = 0.3\,V/D$, where $V$ is the transverse wind speed and $D$ the aperture diameter. This is not confirmed in our data, as wind speeds were no more than a few tens of m/s and our beam diameter was about 34 mm. The corresponding theoretical transition frequency is significantly lower than the ≈400 Hz visible in Fig. 2. We attribute that discrepancy mainly to the fact that the theoretical calculations in ref. [61] were done for a plane wave impinging on a circular aperture, while our beam is Gaussian and smaller than the receiving aperture, and we note that discrepancies between the theoretical model and experimental measurements have been reported previously (see e.g. Tab. I in ref. [57]).

When the stabilization system is turned on, we see around eight orders of magnitude reduction in phase noise PSD at 1 Hz, down to $2.7 \times 10^{-6}\,\mathrm{rad^2\,Hz^{-1}}$. Having the active tip-tilt terminal engaged appears to offer a slight improvement in phase-stability. At frequencies above roughly 2 kHz, the phase noise performance is limited by the residual phase noise of the laser (this is discussed in Supplementary Note 2), which also affects the unstabilized measurement above ≈10 kHz. At lower frequency (roughly 200 Hz to 2000 Hz), we are most likely limited by the noise floor resulting from the operation of our compensation system when applied to the atmospheric phase noise, as shown in detail in Supplementary Note 2.

The long-term fractional frequency stability of the stabilized signals is shown in Fig. 2b in terms of modified Allan deviation (MDEV). This provides an alternative tool for assessing the

performance of the stabilized optical transfer, with a particular focus on stability at longer time scales. The MDEV, calculated using the same Ettus X300 data, is shown in Fig. 2b both in its raw form (dashed traces), as well as after removal of a quadratic fit in phase (solid traces). The linear and quadratic coefficients were $0.15\,\mathrm{rad\,s^{-1}}$ and $6.1 \times 10^{-7}\,\mathrm{rad\,s^{-2}}$ for the tip-tilt on data (and $0.14\,\mathrm{rad\,s^{-1}}$ and $-1.3 \times 10^{-7}\,\mathrm{rad\,s^{-2}}$ for tip-tilt off). We attribute the linear drift to a known offset (measured as $0.141\,\mathrm{rad\,s^{-1}}$) produced in the Ettus. The residual linear drift after accounting for the Ettus is $<9\,\mathrm{mrad\,s^{-1}}$ and results in a systematic offset of $<1.5\,\mathrm{mHz}$ (or a fractional offset of $<7.5 \times 10^{-18}$). This systematic offset does not impact the transfer stability; however, it would need to be taken into account when calibrating a true optical clock comparison. We further conclude that any residual drift is due to thermally induced variations in the differential optical length change of the uncompensated short (~60 cm) fibers between the laser and first splitters on the transmitter side, and last splitters and photo-diode on the receiver side (refer to Fig. 1). We expect that modest temperature control can decrease the quadratic effect by about an order of magnitude, hence the drift removed stability (solid lines) is likely to reflect the ultimate potential of our method.

The MDEV averages as a combination of $\tau^{-3/2}$ and $\tau^{-1}$ power laws until an integration time of around 20 s, indicating that the dominant noise at short timescales is white phase and flicker phase noise, in agreement with the phase noise PSD. The optimum stability reached when the active tip-tilt control system was turned off is $3.0 \times 10^{-19}$ at 40 s of integration time.

When the active tip-tilt terminal is engaged, a slight improvement in stability is seen for integration times longer

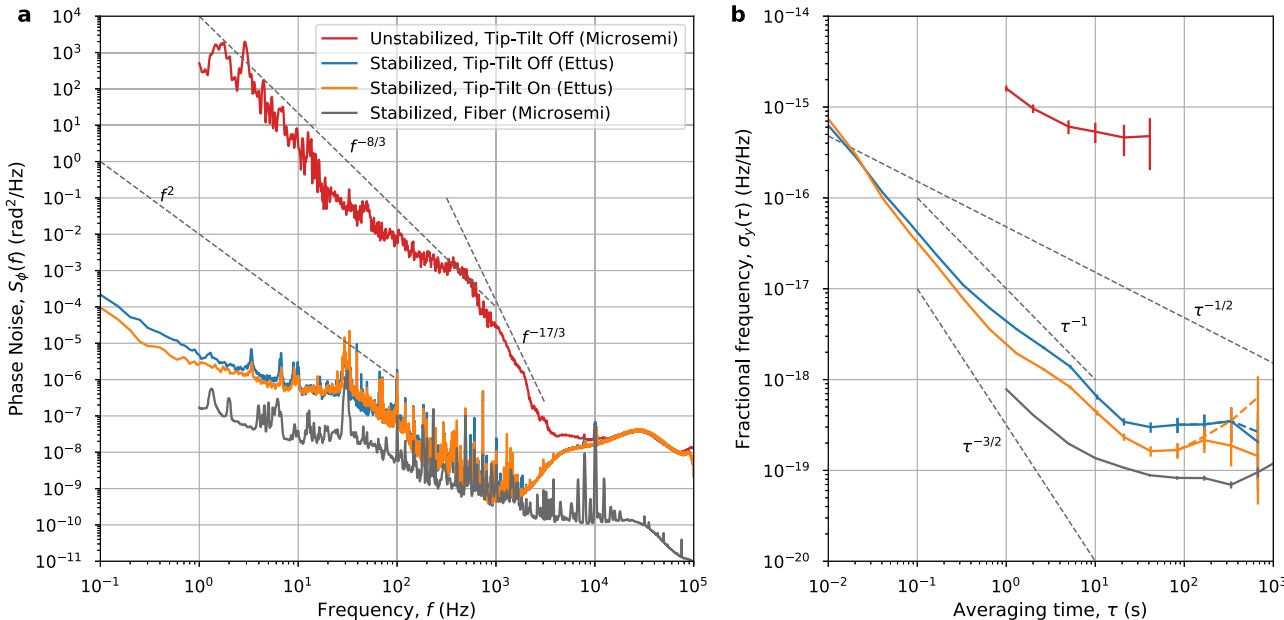

**Fig. 2 Phase and frequency stability of the optical transmission measured at the remote site.** Red trace, free-space link phase stabilization off, tip-tilt active optics off (data from Microsemi); blue trace, free-space link phase stabilization on, tip-tilt active optics off (data from Ettus); orange trace, free-space link phase stabilization on, tip-tilt active optics on (data from Ettus); and gray trace, system noise floor with both phase stabilization systems transmitting over parallel optical fiber (data from Microsemi). **a** Power spectral density of the phase noise ($S_\phi(f)$) after transmission. **b** Fractional frequency stability presented as modified Allan deviation ($\sigma_y(\tau)$). The dashed traces are calculated from raw data; the solid traces are calculated from data with quadratic drift removed; and the error bars represent a standard fractional frequency measurement confidence interval set at $\pm\sigma_y(\tau)/\sqrt{N}$, where $N$ is the number of phase measurements. On both plots, the black dashed lines show key gradients of interest.

than 0.02 s (consistent with the phase noise PSD), and the transfer is made more robust. This results in a fractional frequency stability less than $7 \times 10^{-19}$ for integration times longer than 10 s, with an optimum stability of $1.6 \times 10^{-19}$ achieved at 40 s of integration. This is a factor of two improvement over the case without active tip-tilt control.

At longer timescales, the stability does not integrate down further. This is likely due to long-term residual temperature fluctuations in the local and remote sites affecting the uncompensated parts of the two links, as discussed above, and observed in ref. [12]. With better thermal regulation, the fractional frequency stability is expected to continue averaging down to a lower limit.

The minimum absolute fiber-to-fiber power loss achieved for the one-way transmission was ~12 dB, though this would quickly degrade with poor alignment. The two free-space beam splitters in the optical terminal account for 6–7 dB of the loss, and the remaining is attributed to coupling losses, imperfect alignment, and atmospheric effects.

During operation, the relative power of the optical signal received by the remote site was recorded in order to measure the atmospheric induced fluctuations encountered during a one-way pass of the free-space link. Immediately after the active terminal, a fiber splitter was used to send a small portion of the received signal to a fiber-photo-detector with a linear response to the received optical power. The response of this detector was then digitized at 4 kHz.

Figure 3 shows the frequency domain power of the received power fluctuations. Without the active tip-tilt terminal engaged, the power fluctuations drop as roughly $f^{-2}$ at low frequency and $f^{-3/2}$ beyond a few Hz. The tip-tilt active optical terminal improves the stability at frequencies below 4 Hz, with over two orders of magnitude reduction in power fluctuations at 0.1 Hz. The tip-tilt servo bump at ~7 Hz is clearly visible. Beyond that

bump, there is not a significant difference between having the tip-tilt compensation on or off, as expected. It is interesting to note that the ~4 Hz crossing point roughly matches the frequency at which the phase noise PSD in Fig. 2 starts improving for tip-tilt on (with respect to off), confirming that the phase noise reduction is related to the reduction in power fluctuations. This also implies that better performance of the active optics system (hence lower power fluctuations) is likely to lead to lower phase noise. The tip-tilt system was based on a commercially available unit and the low bandwidth of the system is due to the low gain setting necessary to mitigate some artifacts in the control system firmware (discussed further in "Methods").

The time domain plots and the histogram provide additional representations of the effect of the active terminal. Without tip-tilt, the optical power fluctuates significantly, and at around 100 s, there is a step change in the received power. This was likely due to mechanical movement of the optical terminal, such as mechanisms in the telescope mount suddenly slipping. This step in power can also be seen in the bi-modal distribution of the histogram. When the tip-tilt actuation was activated, step behavior like this was not observed.

Taking the bi-modal feature due to movement of the optical terminal into account, the histograms for both the tip-tilt on and off cases exhibit a log-normal distribution, as is expected of power fluctuations caused by turbulence-induced beam wander. The case with the tip-tilt system engaged shows a much narrower distribution in received power, indicating more constant optical power levels delivered to the phase stabilization system. This indicates that the tip-tilt active optics are effective at suppressing power fluctuations caused by atmospheric turbulence or movement of the terminals.

For clarity, the optical power time series traces shown in Fig. 3 are normalized to their own average power level. With the tip-tilt

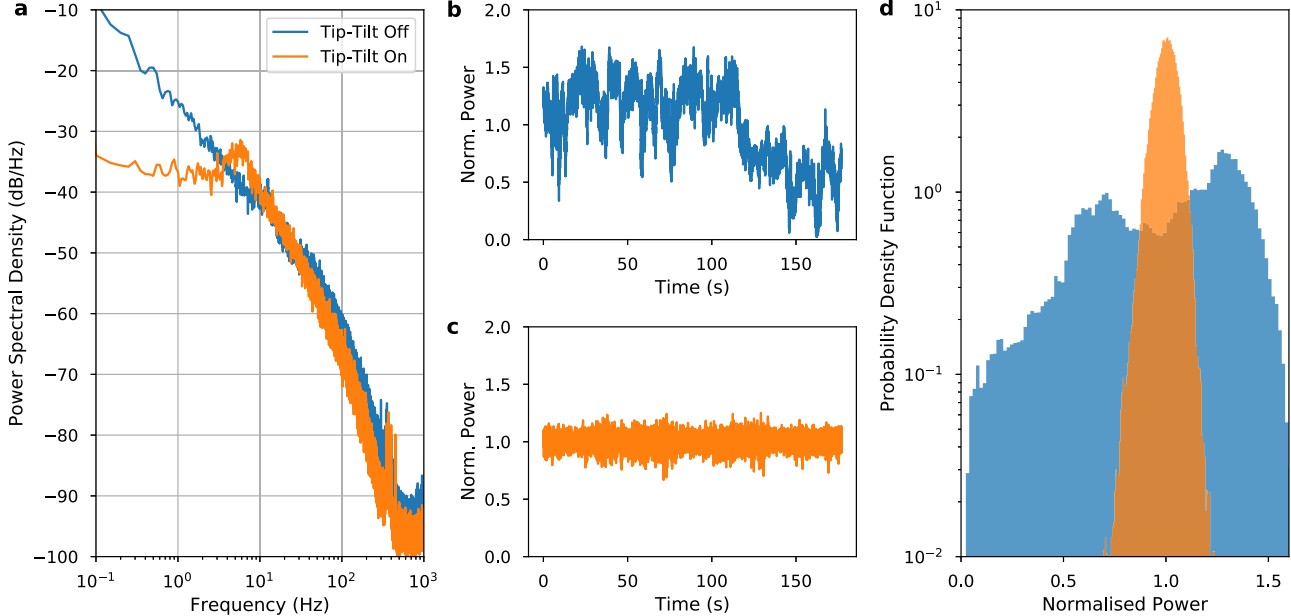

**Fig. 3 Normalized power ($P/\langle P \rangle$) of the free-space optical signal received at the remote site with over 3 min.** Blue trace, tip-tilt active optics off; and orange trace, tip-tilt active optics on. **a** Power spectral density of the received power. **b** Time series of received power with tip-tilt active optics off. **c** Time series of received power with tip-tilt active optics on. **d** Histogram of the normalized received power values.

system on, the average optical power received at the remote site was 2.4 times higher than the average power level when the tip-tilt system was off. Critically, with the tip-tilt system on, the optical power does not make significant excursions into lower power values, greatly reducing the chance of a cycle slip in the phase stabilization system.

## Discussion

The transfer of stable optical frequency reference signals over free-space is of particular interest to applications involving ad-hoc transmissions between mobile sites. A specific example of interest is chronometric geodesy[28–34], where frequency comparisons with a mobile optical atomic clock at different positions over the region of interest provide a direct measurement of the gravitational red-shift caused by changes in gravitational field and height. The requirements are that the transfer system provide sufficiently stable optical transmission so that the uncertainty of the frequency comparison is limited by the uncertainty of the optical atomic clocks themselves, be physically robust and portable, and be light and small enough to allow for easy and rapid set up of the terminals in different locations.

The stabilities of the best lab-based optical atomic clocks are approaching $10^{-18}$ for averaging times on the order of $10^3$ s[2,28,62–65]. Bothwell et al.[63] achieve a stability of $4.8 \times 10^{-17}/\sqrt{\tau}$ (represented by the $\tau^{-1/2}$ gradient line in Fig. 2), averaging down to a final systematic uncertainty-dominated stability of $2 \times 10^{-18}$ within 10 min. The stability demonstrated using the system described in this paper surpasses this stability by more than an order of magnitude, ensuring that frequency comparison between optical clocks over a turbulent free-space channel such as this will not be limited by the performance of the phase-stabilized link.

Our system is also designed to be physically robust and portable (as shown in Fig. 1). The optical terminals are securely built within a steel enclosure that provides protection during transport and while the link is operational. Each terminal has a mass of 14.5 kg, and is 49 cm wide, 24 cm deep, and 18 cm high. The optical fiber-based phase stabilization systems are built within 19"

rack-mount steel and aluminum enclosures. The transmitter module is 2U high, 34 cm deep, and has a mass of 11.6 kg; while the receiver module is 1U high, 25 cm deep, with a mass of 5.9 kg. It should be noted that there is scope for significant reduction in size and weight through the use of custom-engineered components. The robustness of the terminals was demonstrated by the fact that they were successfully shipped, via conventional couriers, from Perth, Australia to Toulouse, France without damage or misalignment of the optics. One of the terminals was installed in a telescope dome for the duration of the 2-week trial period, while the other terminal was set up on an open rooftop and was removed and reset every day. A co-aligned visible guide laser and a simple mount scanning algorithm were used to set the link alignment each morning, and initial alignment could be completed within ~15 min. Throughout the day, the link alignment would occasionally need to be re-optimized. We have since commenced the development of a co-aligned camera with a machine vision imaging system to automate link acquisition to under a minute.

The long-term operation of the system was limited by the performance of the tip-tilt active optics due to the relatively low sensitivity of the Quadrant Photo Detector (QPD), as discussed further in "Methods". The lower limit of the QPD's operational detected power range is −10 dBm (0.1 mW), whereas the phase stabilization system is capable of operating with ~−54 dBm (4.0 nW) of light returning to the transmitter unit[66]. Thus, large drops in link power would first affect the QPD, causing the tip-tilt system to lose the link alignment, and resulting in a loss of signal and cycle slips in the phase stabilization system. Additionally, the tip-tilt system lacked the ability to consistently recover from loses in link alignment. This resulted in the link being able to consistently achieve cycle-slip and deep-fade-free operation for time periods on the order of $3 \times 10^3$ s, before the tip-tilt system lost link alignment. For the system to be able to operate over longer periods of time, the tip-tilt system has to be improved to operate with less stringent power requirements and to be able to effectively re-acquire the link.

There are additional challenges associated with extending the link to beyond 265 m, including more severe atmospheric effects

and increased power losses. The more severe atmospheric effects will require higher tip-tilt suppression bandwidth and steering range. This will involve improving the feedback transfer function to more effectively deal with the frequency resonances of the tip-tilt mirror, or replacing the mirror and actuators with alternatives that have higher resonances. The decreased optical power associated with longer links will exacerbate the issues caused by the low sensitivity of the QPD. Our plan to overcome this is to replace the QPD with a more sensitive equivalent and increase the power of the transmitted beam with a high-power amplifier. This should allow operation over longer links, without having to significantly increase the complexity of the active optical terminals.

An alternative method of dealing with the greater power losses associated with longer links would be to increase the size of the apertures. This however introduces other difficulties. If the size of the apertures increases to much larger than the Fried parameter, then higher-order spatial effects of the atmosphere will start to become significant and lead to speckle and scintillation[19]. Complex and expensive adaptive optics would be required to suppress these higher order effects. Simulations, similar to those published by Robert et al.[19], indicate that tip-tilt is sufficient at keeping power fluctuations low for links to a stratospheric platform at a 50 km distance, provided the apertures remain below around 10 cm. This reduces the required complexity of the optics, but at the cost of higher absolute link loss.

While the focus of our research has been terrestrial links between mobile optical atomic clocks, it is worth noting that the compact nature of the demonstrated system may prove useful for future satellite-to-satellite timing links. The significant weight, and power consumption costs associated with satellite instrumentation, lends itself to the simple system demonstrated in this paper. Additionally, the reduction in atmospheric effects associated with satellite-to-satellite transmission may reduce the challenges associated with the active optics. There are however many other challenges associated with creating a coherent satellite-to-satellite link that have not been captured within the experiment described in this paper. For example, the phase stabilization technique will work only with reduced bandwidth due to the longer transmission time, and be affected by large Doppler shifts. While solutions exist (e.g. corrections in post-analysis), a significant amount of system development and experimentation would be required before translating the system to space-based links.

The long-term goal of our collaboration is to work toward a practical system for performing high-precision clock comparisons between mobile atomic clocks for the purposes of chronometric geodesy. This application requires the use of ad-hoc free-space links between mobile optical atomic clocks separated by up to 100 km, and without necessarily having line of sight. For this extreme application, beyond having to overcome the power and atmospheric challenges mentioned above, an active relay off an airborne platform would be required. The results of this paper represent the first steps toward this ambitious long-term goal.

## Methods

**Phase stabilization system.** Two phase stabilization systems, with very similar architectures, are used to stabilize the free-space and fiber paths. For simplicity, we assume negligible propagation delay and consider only link noise in this section, but these assumptions are revisited in Supplementary Note 2. Equivalent variables relating to the free-space and fiber stabilization systems are identified by superscripts of fs and fb, respectively.

The stabilization systems are based on the imbalanced Michelson interferometer design developed by Ma et al.[67,68], where the long arm of the interferometer is sent over the link and the short arm is reflected by a Faraday mirror to provide an optical frequency reference. The frequency of the outgoing optical signal is shifted by a transmission acousto-optic modulator (AOM) with a nominal frequency ($\nu_{tr}$) which may be varied ($\Delta\nu_{tr}$). The shifted optical signal is then sent over the link.

In the free-space system, the signal is passed through the active terminal described below and launched over the free-space link. The signal then reaches the remote site after picking up link phase noise caused mainly by atmospheric turbulence ($\delta\nu^{fs}$). This optical signal is received by a second active optical terminal and passed through a bi-directional optical amplifier to offset the signal power lost during transmission.

In the fiber system, the signal is passed through an underground fiber running between the two sites. The transmitted signal picks up link noise due to mechanical and thermal fluctuations along this fiber ($\delta\nu^{fb}$).

At the remote site, each stabilization system passes their received signal through an anti-reflection AOM ($\nu_{ar}$), before outputting half the signal to the end user ($\nu_{out}$). The output at the remote site of the free-space stabilization system is given by

$$\nu_{out}^{fs} = \nu_L + \nu_{tr}^{fs} + \Delta\nu_{tr}^{fs} + \nu_{ar}^{fs} + \delta\nu^{fs} , \qquad (1)$$

where $\nu_L$ is the laser frequency, while the output of the fiber stabilization system is given by

$$\nu_{out}^{fb} = \nu_L + \nu_{tr}^{fb} + \Delta\nu_{tr}^{fb} + \nu_{ar}^{fb} + \delta\nu^{fb} . \qquad (2)$$

The two signals are optically beat together at a photodetector and low-pass filtered to produce a down-converted signal,

$$\nu_{meas} = \nu_{out}^{fs} - \nu_{out}^{fb}, \qquad (3)$$

used to measure the relative stability of the optical signals reaching the remote site through free-space and fiber. The residual phase noise from the free-space transmission dominates from the residual phase noise from the fiber transmission over most of the Fourier frequency range. The AOM frequencies were chosen so that the measured beat signal ($\nu_{meas}$) was at a nominal frequency of 1 MHz. An external 10 MHz signal from a hydrogen maser was shared between the two sites via radio frequency (RF) over fiber and provided a common reference for the transmitter oscillators and remote site measurement equipment. As the frequency of the RF reference is seven orders of magnitude lower than the optical signal, the frequency stability of the RF reference will not significantly degrade the phase measurement taken by the remote site measurement equipment.

The other half of the signals reaching the remote site are reflected by Faraday mirrors back through the anti-reflection AOMs and back over the free-space and fiber links. For the free-space link, the return signal also passed back through the bi-directional optical amplifier. At the local site, the signals returning from the fiber and free-space links pass back through their respective transmission AOMs. Each system then performs a self-heterodyne measurement by beating the returned signal against the short arm of the Michelson interferometer on a photodetector. The final electrical beat signal,

$$\nu_{beat} = 2\nu_{tr} + 2\Delta\nu_{tr} + 2\nu_{ar} + 2\delta\nu, \qquad (4)$$

now contains information about the phase noise picked up during the transmission over the link.

This signal is then mixed with a local oscillator of frequency ($2\nu_{tr} + 2\nu_{ar}$) and low-pass filtered in order to extract a DC error signal,

$$\nu_{dc} = 2\Delta\nu_{tr} + 2\delta\nu, \qquad (5)$$

for the phase-locked loop (PLL) that stabilizes the transmission frequency.

The PLL then controls the frequency of the transmission AOM in order to drive this error signal to zero, such that $\Delta\nu_{tr} = -\delta\nu$. This has the effect of suppressing the link phase noise from the free-space (Eq. 1) and fiber (Eq. 2) output signals.

**Active optical terminals.** The active terminals (Fig. 1) used at each end of the free-space link were reciprocal and identical. The optical signal is passed through a fiber to free-space collimator with a $1/e^2$ radius of 1.12 mm. This is then passed through a 50–50 beam splitter (BS). Half the optical signal is sent to a beam-dump, and the other half is sent to a 15:1 Galilean beam expander (GBE) with a 48 mm clear aperture.

The signal from the GBE is reflected off a 50 mm flat mirror with active piezo-electric actuators and launched over the free-space link with a $1/e^2$ radius and divergence of approximately 16.8 mm and 29 μrad, respectively.

The incoming beam is reflected by the active mirror into the GBE. The BS then sends half the incoming light to the free-space-to-fiber collimator, and the other half to a QPD. This QPD is used to detect first-order spatial fluctuations in the incoming beam. The measured fluctuations are passed through a Proportional Integral (PI) controller and used to drive the piezo-electric actuators on the active mirror in order to suppress these fluctuations and keep the incoming beam centered on the QPD. The QPD is positioned so that the optical signal coupled by the collimator into the fiber is maximized when the beam is centered on the QPD.

The QPD and active mirror control system is a commercial off-the-shelf system. The achievable turbulence suppression bandwidth of the system during these tests was limited by the low PI controller gain settings which were necessary to reduce the sensitivity of the tip-tilt system to noise in the QPD when the link optical power dropped below the threshold for effective operation of the QPD. When the optical power dropped below this threshold, the tip-tilt system would attempt to steer to the false beam centroid caused by the detector noise, losing the real beam in the

process. The low gain settings prevented the tip-tilt system from steering too far off target before sufficient optical power was restored.

## Data availability

The data that support the findings of this study are available from the corresponding author, B.P.D.-M., upon reasonable request.

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

## Acknowledgements

This research was supported by the Australian Research Council's Centre of Excellence for Engineered Quantum Systems (EQUS, project ID CE170100009), a UWA Research Collaboration Award, the Programme National GRAM of CNRS/INSU with INP and IN2P3 co-funded by CNES, as well as CS-programme blanc and PhyFOG of Paris Observatory. D.R.G. is supported by a Forrest Research Foundation Fellowship. B.P.D.-M. is supported by an Australian Government Research Training Program (RTP) Scholarship, and a CSIRO Alumni Scholarship in Physics.

## Author contributions

B.P.D.-M., S.W.S., F-X.E., T.L., C.G., D.D., and S.K. built the experimental system. B.P.D.-M., S.W.S., D.R.G., E.S., F.-X.E., T.L., and P.W. ran the measurement campaign. Data analysis and interpretation was performed by B.P.D.-M., D.R.G., E.S., P.W., S.W.S., and M.T. B.P.D.-M. wrote the manuscript with contributions from all authors.

## Competing interests

The authors declare no competing interests.
