## [Peer Review File · Nature Communications]

Reviewer #1 (Remarks to the Author):

The authors in "Point-to-Point Stabilized Optical Frequency Transfer with Active Optics" present the use of robust, compact free-space optical terminals to demonstrate frequency transfer over a ~ 300 m free-space link. The authors demonstrate that their (relatively) simple and compact system has a much lower residual stability than state-of-the-art optical clocks (by over an order magnitude at 100 sec averaging) despite the presence of atmospheric turbulence. I agree with the authors that this approach is likely to be very successful for clock network applications if the distance the free-space link needs to span is on the order of a kilometer, either to connect into a larger fiber network or to connect clocks who are within a few kilometers. I'm less certain about the authors' assertion that they can support 100-km links to a relay in the presence of turbulence. (See detailed comments below.) The application that is not discussed by the authors but that I believe is a natural fit to this approach is long distance satellite-to-satellite links.

The manuscript is clearly written and technically sound. In terms of recommending publication in Nature Communications, this manuscript represents an advance in the field similar to Ref. 8 (H. J. Kang et al, "Free-space transfer of comb-rooted optical frequencies over an 18-km open-air link") which was published in Nature Communications in 2019.

I have provided some detailed comments below.

1. I'm curious if "continuous coherent transmission for up to an hour" means that the authors were able to see zero phase-slips in the relative phase at the remote site between the free-space-transferred and fiber-transferred frequencies.
2. Related to (1), it would be very helpful for the authors to include information about the rate of phase-slips in their phase-noise cancelled signal with their tip/tilt correction on and off even though phase-continuity is not required for only frequency.
3. In this sentence, did authors mean "build on" instead of "lead on": "These efforts lead on from optical timing links developed for timescale comparisons between microwave atomic clocks [47-49]."
4. The authors mention extension to 100-km links to relays for which the turbulence-induced fluctuations will be much stronger (and thus to avoid too many signal fades, the active optics feedback requirements will be greater). Could the authors provide a sentence or two explaining how they plan to handle the greater active optics feedback requirements?
5. If the authors are short on space in the methods section, Section 3.1 could be shrunk as the noise cancellation scheme is pretty standard now over fiber links and the free-space cancellation scheme here mirrors that of the fiber. (This comment should not be taken as a criticism of the nice execution of the phase-noise cancellation presented here.)
6. Could the authors address the possibility of using this technique for future satellite-to-satellite timing links in the Discussion section?
7. Could the authors address the maximum (terrestrial) range over which they could maintain the very flat normalized power of Fig. 3c with only tip/tilt correction, i.e. without requiring larger, expensive adaptive optics systems?
8. This is related to the points above, but in general the manuscript could address the issue of extending beyond the 265-m path demonstrated here in more detail.

Reviewer #2 (Remarks to the Author):

The manuscript "Point-to-Point Stabilized Optical Frequency Transfer with Active Optics" by Benjamin P. Dix-Matthews et al. describes optical frequency transfer over 265 m in a free-space

link outside buildings with the frequency instability better than state-of-the-art optical atomic clocks. The paper describes that optical frequency is transferred from a rooftop of a building to another building in a free-space link with small additional phase noise of $3 \times 10^{-6} \text{ rad}^2 \text{ Hz}^{-1}$ at 1 Hz and fractional frequency instability of 1.6×10^{-19} after 40 s integration by using transportable optical terminal equipped with active tip-tilt optics to suppress beam wander. The obtained instability is much smaller than the previous free-space link experiment by F. R. Giorgetta et al., Nat. Photon. 7, 434 (2013) Ref. [21] and the instability of state-of-the-art optical atomic clocks. The manuscript is interesting and certainly of interest for the community. I therefore recommend publication in Nature Communications with some changes/clarifications to be, however, strongly considered.

I have several comments that should be addressed in the revised version:

The authors describe the requirements for the system such as low size, weight, and power in page 2, but only information on size of a transmitter module and a receiver module is given. The information (size and weight) for an optical terminal should be given.

The description "... in phase noise at 1Hz, down to $1.0 \times 10^{-5} \text{ rad}^2 \text{ Hz}^{-1}$, ..." in page 2 is not consistent with " $3 \times 10^{-6} \text{ rad}^2 \text{ Hz}^{-1}$ at 1 Hz" described in abstract and page 4.

Frequency instability for unstabilized (Tip-Tilt off) case is a helpful information for the community, and therefore should be added in Fig. 2b like Fig. 2a.

In page 4 and supplementary material, the authors compare Microsemi 3120A and Ettus X300 (Software Defined Radio) using a noisy signal from the free-space optical transfer setup, but the comparison using a clean RF signal provided directly by the RF reference seems to be easier and obvious to know the performance of Ettus X300 relative to Microsemi 3120A.

What kind of RF reference is used for the measurement? To my understanding, instabilities of the RF reference used for phase stabilization in Michelson interferometer are cancelled between two links, but the performance of the RF reference affects phase measurement at the remote site.

The authors described that continuous, deep-fade free and coherent transmission was achieved for $3 \times 10^3 \text{ s}$ (~ 1 hour). Since long-term, continuous and robust operation is critical for the practical applications of optical atomic clocks, the authors should mention what happens after $3 \times 10^3 \text{ s}$ or what is limiting the longer-term operation.

The authors describe that the modified Allan deviation (MDEV) in Fig. 2b was calculated after removing linear and quadratic effects of 0.15 rad/s and $6.1 \times 10^{-7} \text{ rad/s}^2$. The linear term of 0.15 rad/s corresponds to the frequency offset of $0.15/(2\pi) = 0.024 \text{ Hz}$ or 1.2×10^{-16} at 1550 nm (193 THz), which is much larger than the obtained instability of 1.6×10^{-19} and instability and uncertainty of state-of-the-art optical atomic clocks of $\sim 10^{-18}$. The quadratic term of $6.1 \times 10^{-7} \text{ rad/s}^2$ results in the frequency shift of $6.1 \times 10^{-7} \times 3 \times 10^3 = 0.0018 \text{ rad/s}$ after 3×10^3 which corresponds to $0.0018/(2\pi) = 0.29 \text{ mHz} = 1.5 \times 10^{-18}$ at 1550 nm (193 THz). This estimation may not be true, but the authors should mention why the linear and quadratic drifts are removed in the analysis or should discuss the linear and quadratic effects.

What do the error bars in Fig. 2b stand for?

What is the optical power loss for one way transfer from the local site to the remote site?

The authors describe that optical terminal on an open rooftop location was removed and reset every day. Since transportability of the system is one of appealing points, it would be attractive to describe the preparation such as how to set the initial beam alignment, the preparation time before starting measurement, and so on.

The supplementary material describes that the refractive index of the fiber is $n = 1.65$. Is this true? To my understanding, typical refractive index of silica fiber is ~ 1.45 .

We would like to thank the reviewers for their thorough and constructive comments. Below we detail point-by-point responses to each of the points raised by the reviewers.

Reviewer #1 (Remarks to the Author):

The authors in “Point-to-Point Stabilized Optical Frequency Transfer with Active Optics” present the use of robust, compact free-space optical terminals to demonstrate frequency transfer over a ~ 300 m free-space link. The authors demonstrate that their (relatively) simple and compact system has a much lower residual stability than state-of-the-art optical clocks (by over an order magnitude at 100 sec averaging) despite the presence of atmospheric turbulence. I agree with the authors that this approach is likely to be very successful for clock network applications if the distance the free-space link needs to span is on the order of a kilometer, either to connect into a larger fiber network or to connect clocks who are within a few kilometers. I’m less certain about the authors’ assertion that they can support 100-km links to a relay in the presence of turbulence. (See detailed comments below.) The application that is not discussed by the authors but that I believe is a natural fit to this approach is long distance satellite-to-satellite links.

The manuscript is clearly written and technically sound. In terms of recommending publication in Nature Communications, this manuscript represents an advance in the field similar to Ref. 8 (H. J. Kang et al, “Free-space transfer of comb-rooted optical frequencies over an 18-km open-air link”) which was published in Nature Communications in 2019.

I have provided some detailed comments below.

1. I’m curious if “continuous coherent transmission for up to an hour” means that the authors were able to see zero phase-slips in the relative phase at the remote site between the free-space-transferred and fiber-transferred frequencies.

Yes, over the measurement periods presented in this paper we achieved phase coherent transmission with zero phase-slips between the fibre and free-space links. The following sentences have been changed to reflect this.

Line 22: “The active optical terminals enable continuous, cycle-slip free, coherent transmission over periods longer than an hour.”

Line 71: “The active spatial stabilization used at each terminal is effective at suppressing beam wander caused by the atmospheric turbulence, allowing continuous, cycle-slip and deep-fade free, coherent transmission over periods longer than an hour.”

2. Related to (1), it would be very helpful for the authors to include information about the rate of phase-slips in their phase-noise cancelled signal with their tip/tilt correction on and off even though phase-continuity is not required for only frequency.

We have very little data on the rate of cycle slips, predominantly due to the fact that they were low. We focused on taking measurements when the atmospheric turbulence was at a minimum, as the tip-tilt system struggled to lock at higher turbulences, which resulted in phase-slips being at a minimum for both tip-tilt on and off.

We have a limited amount of time series data from a frequency counter which indicates when cycle slips were encountered. Over two 4000s measurements, we encountered 3 phase-slips with the tip-tilt off and zero with the tip-tilt on. While interesting, we don't believe that the small number of cycle slips that we successfully recorded can be used to provide much meaningful information.

3. In this sentence, did authors mean "build on" instead of "lead on": "These efforts lead on from optical timing links developed for timescale comparisons between microwave atomic clocks [47-49]."

The reviewer is correct. The sentence has been changed to the following:

Line 32: "These efforts **build on** optical timing links developed for timescale comparison between microwave atomic clocks..."

4. The authors mention extension to 100-km links to relays for which the turbulence-induced fluctuations will be much stronger (and thus to avoid too many signal fades, the active optics feedback requirements will be greater). Could the authors provide a sentence or two explaining how they plan to handle the greater active optics feedback requirements?

We have included some text (lines 206 to 254) explaining in detail our future plans and how we expect to achieve them. The referee is correct in pointing out the difficulties that lie ahead, and we have tried to express that in the new text.

5. If the authors are short on space in the methods section, Section 3.1 could be shrunk as the noise cancellation scheme is pretty standard now over fiber links and the free-space cancellation scheme here mirrors that of the fiber. (This comment should not be taken as a criticism of the nice execution of the phase-noise cancellation presented here.)

While we agree that the scheme being used is pretty standard and well understood in the field, we believe that the detailed explanation we have provided will make the paper more accessible for the wider audience of Nature Communications readership.

6. Could the authors address the possibility of using this technique for future satellite-to-satellite timing links in the Discussion section?

We have included the following paragraph discussing this point.

Line 239: "While the focus of our research has been terrestrial links between mobile optical atomic clocks, it is worth noting that the compact nature of the demonstrated system may prove useful for future satellite-to-satellite timing links. The significant weight, and power consumption costs associated with satellite instrumentation lends itself to the simple system demonstrated in this paper. Additionally, the reduction in atmospheric effects associated with satellite-to-satellite transmission may reduce the challenges associated with the active optics. There are however many other challenges associated with creating a coherent satellite-to-satellite link that have not been captured within the experiment described in this paper. For example, the phase stabilization technique will work only with reduced bandwidth due to the longer transmission time, and be affected by large Doppler shifts. While solutions exist (e.g. corrections in post-analysis) a significant amount of system development and experimentation would be required before translating the system to space-based links."

7. Could the authors address the maximum (terrestrial) range over which they could maintain the

very flat normalized power of Fig. 3c with only tip/tilt correction, i.e. without requiring larger, expensive adaptive optics systems?

We have added the following paragraph to provide more information about this point.

Line 231: “An alternative method of dealing with the greater power losses associated with longer links would be to increase the size of the apertures. This however introduces other difficulties. If the size of the apertures increases to much larger than the Fried parameter, then higher order spatial effects of the atmosphere will start to become significant and lead to scintillation [19]. Complex and expensive adaptive optics would be required to suppress these higher order effects. Simulations, similar to those published in [19], indicate that tip-tilt is sufficient at keeping power fluctuations low for links to a stratospheric platform at 50 km distance, provided the apertures remain below around 10 cm. This reduces the required complexity of the optics, but at the cost of higher absolute link loss.”

8. This is related to the points above, but in general the manuscript could address the issue of extending beyond the 265-m path demonstrated here in more detail.

We have added the following paragraph to discuss this point.

Line 222: “There are additional challenges associated with extending the link to beyond 265 m, including more severe atmospheric effects and increased power losses. The more severe atmospheric effects will require higher tip-tilt suppression bandwidth and steering range. This will involve improving the feedback transfer function to more effectively deal with the frequency resonances of the tip-tilt mirror, or replacing the mirror and actuators with alternatives that have higher resonances. The decreased optical power associated with longer links will exacerbate the issues caused by the low sensitivity of the QPD. Our plan to overcome this is to replace the QPD with a more sensitive equivalent and increase the power of the transmitted beam with a high power amplifier. This should allow operation over longer links, without having to significantly increase the complexity of the active optical terminals.”

Reviewer #2 (Remarks to the Author):

The manuscript “Point-to-Point Stabilized Optical Frequency Transfer with Active Optics” by Benjamin P. Dix-Matthews et al. describes optical frequency transfer over 265 m in a free-space link outside buildings with the frequency instability better than state-of-the-art optical atomic clocks. The paper describes that optical frequency is transferred from a rooftop of a building to another building in a free-space link with small additional phase noise of $3 \times 10^{-6} \text{ rad}^2 \text{ Hz}^{-1}$ at 1 Hz and fractional frequency instability of 1.6×10^{-19} after 40 s integration by using transportable optical terminal equipped with active tip-tilt optics to suppress beam wander. The obtained instability is much smaller than the previous free-space link experiment by F. R. Giorgetta et al., Nat. Photon. 7, 434 (2013) Ref. [21] and the instability of state-of-the-art optical atomic clocks. The manuscript is interesting and certainly of interest for the community. I therefore recommend publication in Nature Communications with some changes/clarifications to be, however, strongly considered.

I have several comments that should be addressed in the revised version:

The authors describe the requirements for the system such as low size, weight, and power in page 2,

but only information on size of a transmitter module and a receiver module is given. The information (size and weight) for an optical terminal should be given.

We agree that this information should be given. We have added the following paragraph to provide this information.

Line 198: “Each terminal has a mass of 14.5 kg, and is 49 cm wide, 24 cm deep, and 18 cm high. The optical fiber-based phase-stabilization systems are also built within 19" rack-mount steel and aluminum enclosures. The transmitter module is 2U high, 34 cm deep, and has a mass of 11.6 kg; while the receiver module is 1U high, 25 cm deep, with a mass of 5.9 kg. It should be noted that there is scope for significant reduction in size and weight through the use of custom-engineered components.”

The description “... in phase noise at 1Hz, down to $1.0 \times 10^{-5} \text{ rad}^2 \text{ Hz}^{-1}$, ...” in page 2 is not consistent with “ $3 \times 10^{-6} \text{ rad}^2 \text{ Hz}^{-1}$ at 1 Hz” described in abstract and page 4.

We thank the reviewer for noticing this typographical error. The corresponding entries have been updated, and they have been quoted to two significant figures to ensure consistency with MDEV.

Line 23: “We achieve residual instabilities of $2.7 \times 10^{-6} \text{ rad}^2 \text{ Hz}^{-1}$ at 1 Hz, and an ultimate fractional frequency stability of 1.6×10^{-19} after 40 s of integration...”

Line 69: “The phase-stabilized free-space optical transfer exhibits an 80 dB improvement in phase noise at 1 Hz, down to $2.7 \times 10^{-6} \text{ rad}^2 \text{ Hz}^{-1}$, compared to the unstabilized optical transmission.”

Line 114: “When the stabilization system is turned on, we see around eight orders of magnitude reduction in phase noise PSD at 1 Hz, down to $2.7 \times 10^{-6} \text{ rad}^2 \text{ Hz}^{-1}$.”

Frequency instability for unstabilized (Tip-Tilt off) case is a helpful information for the community, and therefore should be added in Fig. 2b like Fig. 2a.

We have amended Fig. 2b to include this data. We only have a small sample of around 100s where we were successfully able to operate the Microsemi. The Microsemi is particularly poor at dealing with the large power fluctuations of the free-space link, making it hard to obtain long term measurements using it. Additionally, the phase unstabilized results were obtained using the Microsemi, which during high turbulence tended to be optimistic in its calculation of the MDEV due to heavy internal filtering optimised for the measurement of quartz oscillators.

This is why we switched to using the ETTUS software defined radio. During the experiment campaign we dedicated the majority of our time to obtaining the stabilized measurements, and thus did not have time to acquire the unstabilized results with the ETTUS.

In page 4 and supplementary material, the authors compare Microsemi 3120A and Ettus X300 (Software Defined Radio) using a noisy signal from the free-space optical transfer setup, but the comparison using a clean RF signal provided directly by the RF reference seems to be easier and obvious to know the performance of Ettus X300 relative to Microsemi 3120A.

It is true that the test described will provide information about the noise floors of each system. However, we performed the comparison to confirm that the measurements of a noisy signal (not dominated by the noise of the measurement equipment) for both devices agreed. This indicated that

the comparison between devices was valid. We believe that this confirmation is more valuable for this experiment.

What kind of RF reference is used for the measurement? To my understanding, instabilities of the RF reference used for phase stabilization in Michelson interferometer are cancelled between two links, but the performance of the RF reference affects phase measurement at the remote site.

The following sentence has been amended to clarify this point.

Line 281: “An external 10 MHz signal from a hydrogen maser was shared between the two sites via RF over fiber and provided a common reference for the transmitter oscillators and remote site measurement equipment. As the frequency of the RF reference is seven orders of magnitude lower than the optical carrier, the frequency stability of the reference is unlikely to significantly degrade the phase measurement taken by the remote site measurement equipment.”

The authors described that continuous, deep-fade free and coherent transmission was achieved for 3×10^3 s (~ 1 hour). Since long-term, continuous and robust operation is critical for the practical applications of optical atomic clocks, the authors should mention what happens after 3×10^3 s or what is limiting the longer-term operation.

We agree with the reviewer that this would be valuable information. We have added the following paragraph to discuss this.

Line 211: “The long-term operation of the system was limited by the performance of the tip-tilt active optics due to the relatively low sensitivity of the QPD (Quadrant Photo Detector), as discussed further in Methods. The lower limit of the QPD's operational detected power range is -10 dBm (0.1 mW), whereas the phase stabilization system is capable of operating with ~ -54 dBm (4.0 nW) of light returning to the transmitter unit [66]. Thus, large drops in link power would first affect the QPD, causing the tip-tilt system to lose the link alignment, and resulting in a loss of signal and cycle slips in the phase stabilization system. Additionally, the tip-tilt system lacked the ability to consistently recover from losses in link alignment. This resulted in the link being able to consistently achieve cycle-slip and deep-fade free operation for time periods on the order of 3×10^3 s, before tip-tilt system lost link alignment. For the system to be able to operate over longer periods of time, the tip-tilt system has to be improved to operate with less stringent power requirements and to be able to effectively re-acquire the link.”

The authors describe that the modified Allan deviation (MDEV) in Fig. 2b was calculated after removing linear and quadratic effects of 0.15 rad/s and 6.1×10^{-7} rad/s². The linear term of 0.15 rad/s corresponds to the frequency offset of $0.15/(2 \times \pi) = 0.024$ Hz or 1.2×10^{-16} at 1550 nm (193 THz), which is much larger than the obtained instability of 1.6×10^{-19} and instability and uncertainty of state-of-the-art optical atomic clocks of $\sim 10^{-18}$. The quadratic term of 6.1×10^{-7} rad/s² results in the frequency shift of $6.1 \times 10^{-7} \times 3 \times 10^3 = 0.0018$ rad/s after 3×10^3 which corresponds to $0.0018/(2 \times \pi) = 0.29$ mHz = 1.5×10^{-18} at 1550 nm (193 THz). This estimation may not be true, but the authors should mention why the linear and quadratic drifts are removed in the analysis or should discuss the linear and quadratic effects.

We have amended Fig. 2b to more clearly demonstrate the effect of the quadratic drift. We have also explicitly stated where the linear drift is coming from.

Fig 2, Caption: “For the modified Allan deviation, b, the dashed traces are calculated from raw data; the solid traces are calculated from data with quadratic drift removed...”

Line 122: “The MDEV, calculated using the same Ettus X300 data, is shown in Fig. 2 (b) both in its raw form (dashed lines), as well as after removal of a quadratic fit in phase (solid lines).”

Line 125: “We attribute the linear drift to a known offset (measured as 0.141 rad/s) produced in the Ettus (note, a linear drift does not impact the MDEV values), which is thus an artifact of our measurement system.”

What do the error bars in Fig. 2b stand for?

The following sentence was added at the end of the caption for Fig. 2.

Fig 2, Caption: “... the error bars represent a standard fractional frequency measurement confidence interval set at $\pm\sigma_y(\tau)/\sqrt{N}$, where N is the number of phase measurements.”

What is the optical power loss for one way transfer from the local site to the remote site?

The following two sentences were added in the results section.

Line 145: “The minimum absolute fiber-to-fiber power loss achieved for the one-way transmission was approximately 12 dB, though this would quickly degrade with poor alignment. The two free-space beam splitters in the optical terminal account for 6-7 dB of the loss, and the remaining is attributed to coupling losses, imperfect alignment and atmospheric effects.”

The authors describe that optical terminal on an open rooftop location was removed and reset every day. Since transportability of the system is one of appealing points, it would be attractive to describe the preparation such as how to set the initial beam alignment, the preparation time before starting measurement, and so on.

The following sentence was added to cover this point. The initial link alignment on the first day of tests took several hours due to our reluctance to use a visible guide laser, and that we were relying heavily on the scanning algorithm. On the second day of the campaign we attempted the alignment before dawn, and with a co-aligned visible guide laser. This proved a far more effective strategy and, with a few days practice, we were able to consistently align the link within approximately 15 min.

Line 206: “A co-aligned visible guide laser and a simple mount scanning algorithm were used to set the link alignment each morning, and initial alignment could be completed within approximately 15 min. Throughout the day, the link alignment would occasionally need to be re-optimized. We have since commenced the development of a co-aligned camera with a machine vision imaging system to automate link acquisition to under a minute.”

The supplementary material describes that the refractive index of the fiber is $n = 1.65$. Is this true?

To my understanding, typical refractive index of silica fiber is ~ 1.45 .

We thank the reviewer for noticing this typographical error. The refractive index used in our calculations was ~ 1.45 however this was entered into the manuscript incorrectly. The corresponding sentence has been corrected.

Supp. Line 68: “... (265 m free space + 25 m fibers with refractive index $n=1.45$ between the telescopes and the beam splitters)...”

Reviewer #1 (Remarks to the Author):

The authors have addressed all of the reviewer comments clearly. I recommend publication with no further revisions.

Reviewer #2 (Remarks to the Author):

I thank the authors for their detailed responses to all comments by the referees. I have provided some comments to the revised manuscript and the responses as follows:

In page 4 and supplementary materials: My recommendation on Fig. 1 in supplementary material is to show both the current data and the noise floor of ETTUS measured with a clean RF signal to show that the ETTUS has sufficiently low noise floor and makes a consistent data with the Microsemi, because verifying the ETTUS reliability is one of the key points for the data reliability as the authors described Note 1 in supplementary material.

Line 125: I understand the linear drift of the phase data is an artifact due to the ETTUS, and thus I recommend to explicitly describe that the residual linear phase drift of less than 0.01 rad/s corresponds to the frequency offset of less than 1×10^{-19} .

Figure 2b: Orange dashed lines added in the revised manuscript shows near 6×10^{-19} at 700 s. The authors expect that the quadratic effect can be temperature drift, which may be improved but has not been improved yet, as described in Line 130. Therefore, the conclusion of " 1.6×10^{-19} after 40 s" seems to be optimistic. Judging from the experimental results, "less than 7×10^{-19} after 10 s" or something is thus more appropriate.

Reviewer #1 (Remarks to the Author):

The authors have addressed all of the reviewer comments clearly. I recommend publication with no further revisions.

Reviewer #2 (Remarks to the Author):

I thank the authors for their detailed responses to all comments by the referees. I have provided some comments to the revised manuscript and the responses as follows:

In page 4 and supplementary materials: My recommendation on Fig. 1 in supplementary material is to show both the current data and the noise floor of ETTUS measured with a clean RF signal to show that the ETTUS has sufficiently low noise floor and makes a consistent data with the Microsemi, because verifying the ETTUS reliability is one of the key points for the data reliability as the authors described Note 1 in supplementary material.

We agree that verifying the ETTUS reliability is an important point. We have updated Fig. 1 in the Supplementary Material as suggested by the Reviewer.

Supp. Line 22: "Additionally, the noise floor of the ETTUS was tested by measuring the phase noise of a Rigol DG-4102 synthesizer at 1 MHz. Both the ETTUS and the synthesizer were referenced to the same 10 MHz. The phase noise measured by the ETTUS is shown in Fig. 1. This indicates that the noise floor of the ETTUS was not dominating the measurement."

Supp. Fig 1: "The noise floor of the ETTUS (gray) was obtained by measuring the phase noise of a low noise synthesizer at 1 MHz."

Line 125: I understand the linear drift of the phase data is an artifact due to the ETTUS, and thus I recommend to explicitly describe that the residual linear phase drift of less than 0.01 rad/s corresponds to the frequency offset of less than 1×10^{-19} .

The constant linear phase drift results in a systematic offset in frequency and does not contribute to the fractional frequency stability of the transfer. The residual linear phase drift after accounting for the ETTUS is less than 0.009 rad/s and results in a systematic offset of < 1.5 mHz (or a fractional offset of $< 7.5 \times 10^{-18}$ Hz/Hz). The following sentence has been changed to reflect this.

Line 126: "We attribute the linear drift to a known offset (measured as 0.141 rad s^{-1}) produced in the ETTUS (note, the constant linear drift results in a systematic frequency offset and thus does not impact the transfer stability)..."

Figure 2b: Orange dashed lines added in the revised manuscript shows near 6×10^{-19} at 700 s. The authors expect that the quadratic effect can be temperature drift, which may be improved but has not been improved yet, as described in Line 130. Therefore, the conclusion of " 1.6×10^{-19} after 40 s" seems to be optimistic. Judging from the experimental results, "less than 7×10^{-19} after 10 s" or something is thus more appropriate.

The reviewer makes a good point, that the long-term stability is not expected to be 1.6×10^{-19} for integration times beyond 40 s. We have changed three sentences to clarify that we are quoting the

raw performance at a specific integration time. We have also included the reviewers suggested sentence in the discussion.

Line 23: “We achieve residual instabilities of $2.7 \times 10^{-6} \text{ rad}^2 \text{ Hz}^{-1}$ at 1 Hz, and an optimum fractional frequency stability of 1.6×10^{-19} at 40 s of integration...”

Line 72: “The resulting fractional-frequency stability of the phase-stabilized optical transfer reaches 1.6×10^{-19} with 40 s of integration.”

Line 138: “This results in a fractional frequency stability less than 7×10^{-19} for integration times longer than 10 s, with an optimum stability of 1.6×10^{-19} achieved at 40 s of integration. This is a factor of two improvement over the case without active tip-tilt control.”

Reviewer #2 (Remarks to the Author):

I have just a comment.

Line 126: The residual fractional frequency offset of 7.5×10^{-18} can limit the uncertainty of clock comparisons to this level using this system even if the stability is much smaller than the value. The author should mention the value of the residual fractional frequency offset of 7.5×10^{-18} in the main text in order not to mislead readers.

Reviewer #2 (Remarks to the Author):

I have just a comment.

Line 126: The residual fractional frequency offset of 7.5×10^{-18} can limit the uncertainty of clock comparisons to this level using this system even if the stability is much smaller than the value. The author should mention the value of the residual fractional frequency offset of 7.5×10^{-18} in the main text in order not to mislead readers.

The following sentence has been updated to ensure clarity on this point.

Line 126: We attribute the linear drift to a known offset (measured as 0.141 rad s^{-1}) produced in the Ettus. The residual linear drift after accounting for the Ettus is $<9 \text{ mrad/s}$ and results in a systematic offset of $< 1.5 \text{ mHz}$ (or a fractional offset of $< 7.5 \times 10^{-18} \text{ Hz/Hz}$). This systematic offset does not impact the transfer stability; however, it would need to be taken into account when calibrating a true optical clock comparison.